# Mapping Personality Traits and Gender-Based Stereotypes on Perceived Negotiation Skills

Numrata Moty [1], Manish Putteeraj [1], Jhoti Somanah [1,*] and Krishnee Adnarain-Appadoo [2]

1 School of Health Sciences, University of Technology Mauritius, La Tour Koenig 11134, Mauritius; numrata.moty@gmail.com (N.M.); mputteeraj@utm.ac.mu (M.P.)
2 Faculty of Law and Management, University of Mauritius, Reduit 80837, Mauritius; ka.appadoo@uom.ac.mu
* Correspondence: mjbhugowandeen@utm.ac.mu

**Abstract:** Implementing effective dispute resolution strategies such as negotiation has proved to be quite effective whenever there is a divergence of interest between two conflicting groups. This study aims to see if gender-based stereotypes or specific personality traits can positively or negatively influence negotiation skills in an attempt to improve the negotiation process, whereby individuals could be trained to adopt specific behaviors to obtain more favorable negotiation results. Using the expectancy violation theory (EVT) to analyze how individuals respond to unanticipated violations of social norms and expectations whilst negotiating, a quantitative study was carried out among legal officers working in private, public, and parastatal organizations in Mauritius. The sample size, 270, was calculated based on a population size of 899 as per records of the Mauritius Bar Council. The results demonstrate most legal officers were equipped with good negotiation skills, with no significant difference between males and females (U = 1138.50, $p > 0.05$), while a high level of neuroticism was indicative of poor negotiation skills ($\tau b = -0.167$, $p > 0.05$). These findings demonstrate that participants agreed that their negotiation skills were influenced in gender-dominated meetings which align with the principles of the EVT, the violation of certain expected negotiation behavior based on gender impact negotiation outcomes. Since negotiation skills were significantly associated with negotiation outcomes ($\chi2(4) = 37.963$, $p < 0.05$), this provides pointers to businesses on how to improve and optimize negotiation outcomes by choosing a negotiator with the most apt personality traits.

**Keywords:** expectancy violation theory; gender-based stereotypes; negotiation; personality traits





## 1. Introduction

Negotiation pervades our everyday lives, be it purchasing a plot, sealing a deal at work, bargaining over what to eat [1], or even participating in job interviews, where most employment conditions are customized [2]. In the legal corporate context, negotiation is a complex communication-based activity that incorporates arguments, persuasion, and the exchange of information [1]. In agreement, Fisher [3] argues that "most legal problems are not settled through legislative or judicial action but by negotiation" while Olson [4] suggests that negotiation is present in almost every part of the legal world, that is, all legal officers at some point in their career make use of negotiation through "retainer agreement, partnership agreements, contracts, lawsuit settlement, discovery schedules". Hence, effective negotiation skills are paramount, along with communication skills. The ability to develop contacts, procure resources, conclude difficult agreements, and efficiently manage conflicts illustrates the importance of good negotiation skills [5]. Negotiation is also critical for mergers and acquisitions which [6] bring in more opportunities [7]. However, acquisitions are largely dependent on effective negotiation skills whereby information is shared among buyers and sellers; concessions and compromises are made to reach a mutually profitable agreement [8]. Pruitt and Carnevale [9] claim that the outcomes of negotiation can be influenced by the strategies and tactics adopted by negotiators.

Negotiation behavior is also governed by the negotiator's role in his or her organization, the negotiation style adopted, and the various aspects of the relationship between both parties [10]. Many schools of thought believe that negotiation skills should be compulsory in legal practice as experiential learning [11].

### 1.1. The Role of Personality Traits in Negotiation Skills

Substantial associations have been noted between personality traits and negotiation outcomes as negotiation inherently consists of dealing with the intricacies of human behavior and emotions [12]. Personality is mostly studied in regard to the Big Five [13] which is also known as the Five-Factor Model (FFM) [14]. The FFM categorizes personality traits into five main domains: Openness to Experience, Conscientiousness, Extraversion, Agreeableness, and Neuroticism (OCEAN) [15]. The FFM postulates that individuals have one predominant personality trait that further influences their behaviors [16,17]; an assessment of personality traits may improve negotiating behaviors [18]. For instance, Barry and Friedman [19] validate significant relationships between bargaining success and traits such as agreeableness and extraversion. Similarly, Yiu and Lee [20] demonstrate how openness to experience, conscientiousness, and extraversion can facilitate the negotiation process, hence producing positive results. Morris and Larrick [21] further postulate that negotiating performance can be influenced by the perception of their counterpart's personality, including their agreeableness and level of cooperation, while Spector [22] notes that more than persuasion, personality traits determine the use of specific negotiation strategies as well as negotiation outcomes. Falcão and Saraiva [23] explain how understanding personality traits can help negotiators adapt their behaviors to both distributive and integrative negotiations and possibly improve the negotiation outcomes.

### 1.2. Gender-Based Stereotypes in Negotiation

Gender stereotypes are commonly observed in negotiation. Men, for example, are stereotyped as fiercely competitive, manipulative, win–lose negotiators who seek to defeat their opponents, while female negotiators are presumed to be more agreeable, win–win negotiators who attempt to preserve good relationships by optimizing the combined return obtained by negotiating parties [14,24]. If these stereotypical presumptions are to be believed, then the expectancy that male lawyers or corporate workers shall achieve better results when negotiating as opposed to females is created. Kolb [25] studies the impact of gender stereotypes in negotiation through ambivalent sexism, that is, women who challenge male dominance are treated with hostile attitudes, while those who conform are treated with more benevolent attitudes [26,27]. Unfortunately, in many cases, women tend to internalize these stereotypical beliefs leading to silencing or self-erasure [28,29] causing women to become "more anxious and less willing to negotiate" [25,30]. Given the prominence of negotiation in the legal field, the negotiation process should be as smooth as possible. Nonetheless, gender seems to influence negotiation outcomes; for instance, the gender of the negotiation counterpart was observed to lead to differentiating negotiating outcomes [31]. Unfortunately, it has also been noted that women tend to perform better in same-sex negotiation compared to opposite-sex negotiation while no significant differences were observed for men, pointing to the detrimental effect of gender in negotiation [32].

Negotiation outcomes were mostly one-sided to the high-powered negotiators when stereotypical masculine traits were implicated, while when stereotypical feminine traits were concerned, win–win outcomes were observed. In contrast, it is reported that the "performance in mixed-gender negotiations is strongly affected by the cognitions and motivations that negotiators bring to the bargaining table" [33]. It can be argued that the masculine interpretation of negotiating entails agency and aggressiveness, while understanding human behavior, recognizing nonverbal clues, and developing trust, which are also seen as effective negotiating, are more in line with the traditional female stereotype [33,34]. Consequently, it is believed that males are more efficient in "claiming value" and "creating value" negotiations while females are more apt in negotiating for peaceful

solutions in contentious circumstances [35]. Similarly, Kolb [25] observes that compensation-related negotiations are mostly led by males, while these negotiations are avidly avoided by women [36]. Unfortunately, in many cases, women are observed to be overwhelmed trying to strike the perfect balance between being nice and efficient leaders [37], or in the negotiating jargon, being competitive or accommodating [38]. Thus, societal perceptions of gender roles are not only mirrored in negotiation but also have concrete repercussions for what is negotiable, how problems are presented, bargainers' legitimacy to negotiate over them, and the conceivable results [39].

### 1.3. Expectancy Violation Theory

Perceived negotiation skills were studied using the expectancy violation theory (EVT), which postulates that if the behaviors of negotiators counter the gender-specific beliefs, negative expectancy violations generating backlash and negatively affecting the negotiators' outcomes can be observed [2,40]. Kulik and Olekalns [2] observe that gender-incongruent behavior is challenging for female negotiators because of stereotyped gender-specific expectations in the negotiation process. Interestingly, the contrary is observed for male negotiators, that is, when males counter gender-specific beliefs of being ruthless, it is considered a positive expectancy violation, even being termed a pleasant surprise by opponents [2,40,41]. Accordingly, Heilman and Chen [41] proposed that the altruistic performance of men and women is received differently in the workplace, with men's performance being more favorably evaluated and recommended compared to women's performance.

### 1.4. Significance of Study

Negotiation skills are essential for business relationships [42] as they enhance critical thinking and help develop strong communication skills [43,44]. Putnam and Wilson [45] stated that since conflicts are inevitable in any organization, finding effective ways to manage conflict can prevent further misunderstandings. Considering that Mauritius consists of several corporate firms, this study aims to identify any potential differentials in social power and, accordingly, aid in tackling gender disparities and inequality within institutions and among men and women. The main objectives of this study consist of investigating whether each gender has a particular predominant trait as per the FFM and assessing the individual and joint influence of personality traits and gender stereotypes, if any, on perceived negotiation skills using the EVT. For decades, attempts to balance out gender inequality in the corporate world have been ongoing. Nonetheless, gender differences remain prominent, which can be further observed in negotiation outcomes. Hence, this study seeks to understand differences in perceived negotiation skills through the lenses of gender stereotypes and personality traits. The EVT has been adopted to shed more light on the influence of positive violations and negative violations in the negotiation context.

## 2. Materials and Methods

### 2.1. Study Population

A stratified quantitative survey was used in this study. Considering the strict movement restrictions mandated by the Mauritian Government due to the COVID-19 pandemic, a quantitative study was chosen as it implied a wider administration of the questionnaire. The questionnaire was administered through an online medium targeting legal officers from different sectors in Mauritius, i.e., private, public, and parastatal organizations of Mauritius. The inclusion criterion of the study population was being a legal officer having at least six months of experience in the negotiation field. Furthermore, given the ratio of barristers to attorneys (5:1) within the Mauritian legal system. The sample size, 270, was calculated using Slovin's formula accounting for a 95% confidence interval and a 5% margin of error, based on a population size of 899 as per records of the Mauritius Bar Council. It was estimated that at least 108 barristers would be required, accounting for a response distribution of 50%. Given the plurality of the roles undertaken by legal officers,

segregation according to distinct roles was minimized, while the focus was mainly on precursor negotiation experience.

A pilot test was carried out among 5 legal officers in a public organization to measure the internal validity of the questionnaire through the elimination of any possible ambiguities. No significant issues were identified during the pilot testing, and the questionnaire was deemed adequate for administration. The self-administered questionnaire link was emailed to legal officers, namely counsels, attorneys, compliance officers, and legal executives among others, over a period of five months. A total of 108 responses was collected, which was indicative of a response rate of 40%.

### 2.2. Research Instrument and Design

Questionnaire items were formulated based on the existing pool of literature and documented validated scales related to the main variables as illustrated in Table 1. Databases were screened for original articles using keywords such as gender roles, personality traits, negotiation, and gender-based stereotypes.

**Table 1.** Internal consistency of questionnaire variables.

| Variable | Description | $\alpha$ |
|---|---|---|
| Personality traits | Personality traits were measured using the Mini-IPIP, which is a shortened version of the 50-item International Personality Item Pool (IPIP) [46,47], focused on assessing predominating personality traits. The scale was based on a 5-point Likert scale (1 = very inaccurate to 5 = very accurate). Items 6, 7, 8, 9, 10, 15, 16, 17, 18, 19, and 20 were reverse-scored. A high mean value on a particular trait was indicative of the predominance of that trait label. For instance, a high mean score for Extraversion would tag it as the predominating personality trait. | 0.824 |
| Gender-based stereotypes | This section consists of the Women As Managers Scale (WAMS) [48] as well as five closed-ended questions which were formulated and adapted for this study to assess gender-based stereotypes in the workplace more particularly towards women in management positions. A 7-point Likert scale was used (1 = strongly disagree to 7 = strongly agree). Items 2, 4, 5, 8, 9, 10, 11, 13, 14 and 19 were reverse-scored. A high mean score was indicative of high gender-based stereotypes towards women in the workplace. | 0.882 |
| Negotiation skills | This consisted of the Negotiation Skills Questionnaire by Cook [49] and 3 closed-ended questions which were formulated and adapted for this study. This section focused on assessing the negotiation skills of the participants. Scores obtained for items 1, 2, 3, 6, 12, 13, 14, 18, and 20 were reverse-scored; for item 7, if participants answered C, then it was scored as 5 instead of 3. A higher computed score is indicative of good negotiation skills. | 0.873 |

## 2.3. Data Analysis

Data were analyzed using SPSS version 21.0. Descriptive statistics was used for the inherent characterization of the participants inclusive of demographic information such as profession, gender, and years of experience in negotiation; weighted means were also used for the scale computation. The normality of data was assessed using the Shapiro–Wilk test, and the inferential analysis was adapted accordingly. The Mann–Whitney U test was used to identify gender-based differences and personality traits and assess the different levels of negotiation proficiency against gender-based stereotypes and personality traits. A Kendall's tau-b correlational analysis was undertaken to run comparative analyses between the negotiation skills and personality traits, while associations between negotiation skills and variables of interest such as profession, years of experience, and gender were assessed using chi-square tests. A multiple regression analysis was also used to assess the principles of the EVT. Statistical analyses were reported at a significance level of $p < 0.05$.

## 2.4. Ethical Consideration

Each participant was briefed prior to the start of the survey with respect to the confidential management of the collected data. The survey was accompanied by a cover page whereby participation consent was sought. Information was kept anonymous at all times, and the ethical standards were respected. This study was approved by the Faculty of Law Dissertation Committee, University of Mauritius.

## 3. Results

### 3.1. Demographic Profile of Respondents

The majority of respondents were females (57%) (Supplementary Table S1). The present findings also showed that the legal sector in Mauritius was gaining momentum, with 45.4% aged between 25 and 35 years old, as opposed to seasoned legal professionals aged above 55 years old making up only 8% of the sample. The government-mediated sector accounted for 61.3% of the employees, with a very low percentage having their own practice. Almost equal numbers of males (n = 22) and females (n = 24) were recorded with more than 6 years of experience in negotiation.

### 3.2. Predominance of Personality Traits

The examination of five personality traits was undertaken as per the FFM whereby a high mean value on a particular trait was indicative of that trait label (Supplementary Table S3). For instance, if a high mean score was observed for extraversion, it was indicative of extraversion being the predominant personality trait. The following dominant traits were observed in ranked order from the most to the least prevalent among participants: conscientiousness, agreeableness, openness to experience, extraversion, and finally neuroticism. Gender-based analyses were equally interesting; females were mostly characterized as conscientious (ẋ = 3.83) while males were mainly depicted as bearing the openness to experience (ẋ = 3.45) personality trait. A disparity was also noted in the lowest sub-scale scores: males scored the lowest for neurotic personality, while females scored lowest in two traits, namely neurotic and extraversion. The data assessed through the Likert spectrum were collapsed into three-tiered rating profile, with accurate (A), inaccurate (I), and neutral, and resulted in a higher proportion of participants describing themselves as imaginative (63%), proactive (57.4%), and sympathetic (68.5%), while a split view was observed for mood swings (A vs. I, 30.6% vs. 36.1%) and adopting a lowkey profile (A vs. I, 29.6% vs. 30.6%) (Supplementary Table S4).

Further assessments of predominant personality traits across gender revealed statistically significant differences for conscientiousness (U = 927.5, $p < 0.01$) and neuroticism (U = 852, $p < 0.001$) (Supplementary Table S2). Females scored higher for conscientiousness as opposed to males, endorsing this particular trait as being distinctively assigned to Mauritian females in the legal field, while the opposite was also true for neuroticism, with males scoring significantly lower. The present data support gender differences in specific

traits such as conscientiousness and neuroticism, while other traits that share a certain level of overlap such as openness to experience, extraversion, and agreeableness tend to be distributed equally across gender.

### 3.3. Analysis of Gender-Based Stereotypes

The Women As Managers Scale (WAMS) [48] was used to investigate the prominence of gender-based stereotypes in the workplace. As shown in Supplementary Table S4, most participants believed that both males and females were capable of handling management positions as the majority of the participants were in agreement that gender should not affect equality of opportunity (94.5%). Nonetheless, workplace bias was still observed for item 11, through the illustration of the significant prejudice experienced by Mauritian females in comparison to men due to their biological system and parental responsibilities (89%). It is unfortunate to note that despite advancements to reduce gender inequality in the workplace, women feel threatened that pregnancy might affect their career progression [50]. Interestingly, substantiating Laws's [51] statement of how members of society "have become conditioned to regard the menstrual cycle as the norm and pregnancy as an unnatural event", the present findings show that menstruation was interpreted differently compared to pregnancy given the relatively strong agreement that menstruation is not an element of inferiority (86.1%).

Clustering the WAMS into three measurable factors demonstrated the following: (i) Respondents irrespective of gender embraced women in key hierarchal positions, with 97.3% agreeing on their competence; women, as expected, had a more potent perception of their role and skills (male versus female, 38.68 vs. 66.23; U = 698.5, $p < 0.001$). (ii) Women significantly opposed the notion of women's business acumen and contribution in an enterprise as inferior to men's, with the male counterpart leaning towards a neutral opinion in this case (male versus female, 69.88 vs. 43.09; U = 718.5, $p < 0.001$). (iii) Barriers such as the biological system, emotional cues, and work pressure were not found to deter women in key positions within the workplace, although the strength of rejection with respect to the gendered barriers was not as potent as compared to the other factors (male versus female, 63.24 vs. 48.02; U = 1024, $p < 0.05$). The present findings demonstrate a perception alignment across gender for women to be considered equal in management positions.

A uniform disagreement was observed for items that were mostly regarded as prejudicial to women such that participants disagreed that women could not learn mathematical skills (77.8%), were not ambitious enough (84.3%), were not assertive (81.5%), were not competitive enough (80.5%), and could not be aggressive (71.2%). Although the responses were most likely associated with the entirety of female participants (57%), a number of male participants also opposed the negative statements, aligning with the opposite gender. Likewise, the majority of male and female participants were on the same wavelength when it came to gender stereotypes, with a potent disagreement against views such as jobs (88%) and emotions (95.4%) being gender-specific. Nonetheless, it is important to note that the majority of females felt that they were judged/shamed for not being feminine or masculine enough (n = 42), possibly validating the negative perception of feminine legal officers in court [52] (Supplementary Table S5).

### 3.4. Factors Influencing Negotiation Skills and Negotiation Outcomes

The negotiation skills of participants were assessed based on how they would perform in specific negotiation scenarios. The majority of participants emphasized the need to clarify when communication is unclear (item 14), negotiating with anyone regardless of job title (item 19), and negotiation as a beneficial process to both parties, which are synonymous with good negotiation skills [53]. Interestingly, in accordance with Cook's [49] scoring rules, the study participants were equipped with either excellent or moderate negotiation skills (Supplementary Table S6). No significant relationship was identified between gender and negotiation skills ($\chi 2(1) = 3.107$, $p > 0.05$; Cramer's V = 0.170), in agreement with the almost equal number of males (n = 41) and females (n = 47) with self-reported excellent negotiation

scores. Further analyses to determine the role of demographic variables exerting a potential influence on negotiation skills revealed a significant relationship with (i) profession ($\chi 2(4) = 11.108$, $p < 0.05$; Cramer's V = 0.308) and (ii) the status of a professional organization ($\chi 2(3) = 15.108$, $p < 0.05$; Cramer's V = 0.345), as opposed to age ($\chi 2(3) = 7.385$, $p > 0.05$; Cramer's V = 0.227) and years of experience in negotiation ($\chi 2(3) = 1.990$, $p > 0.05$; Cramer's V = 0.134); the latter could potentially be explained by the relatively uniform distribution of participants in the different groups under the years of negotiation skills except for the 1–5-year segment (n = 41).

As expected, a strong relationship was identified between negotiation skills and negotiation outcomes ($\chi 2(4) = 37.963$, $p < 0.05$; Cramer's V = 620). The present data demonstrate that higher negotiating proficiency led to a more positive outcome ($\tau b = 0.540$, $p < 0.001$), and negotiation skills could potentially be used as an outcome predictor ($\chi 2(1) = 27.41$, $p < 0.05$). However, negotiation skills alone explained only 28.1% of the outcomes, and the observed data did not consolidate the model with a high level of compatibility (goodness of fit, $p < 0.05$). Similar to negotiation skills, negotiation outcomes were not affected by gender on their own ($\chi 2(2) = 3.49015.108$, $p > 0.05$; Cramer's V = 0.180). Therefore, in an attempt to demonstrate the holistic operationalization of the core variables, i.e., skills and stereotypical attributes, the cumulative effect of core variables was analyzed using the principles of the EVT.

*3.5. The Compounding Effect of Gender-Based Stereotypes and Personality Traits on Negotiation Based upon Expectancy Violation Theory (EVT)*

Even though no significant relationship was observed between negotiation skills and predominant personality traits ($\chi 2(4) = 1.45$, $p = 0.835$), segregation within the dimensions of personality traits revealed that only neuroticism was inversely related to negotiation skills such that high neuroticism was indicative of poor negotiation skills ($\tau b = -0.167$, $p < 0.05$). A gender-based characterization of predominant personality traits over proficiency in negotiation skills highlighted conscientiousness as the predominant trait with a higher percentage of females in that cluster (28.7%) (Supplementary Table S7), validating the strong relationship between females and conscientiousness, which has been further linked to the strong sense of responsibility of females and better academic performance [54,55].

Further analyses to identify factors affecting negotiation outcomes demonstrated the significant role of predominant personality traits ($\chi 2(8) = 21.057$, $p < 0.05$; Cramer's V = 0.278) as a pressure point, with conscientiousness being related to more positive outcomes (35.2%), closely followed by openness to experience (24.1%) (Supplementary Table S8).

Factoring the element of gender-based stereotypes, the potential effect of each factor on negotiation skills and outcomes was tested, which mostly resulted in non-significant relationships between gender-based stereotypes, with the exception of Factor 2, i.e., "Enterprise features for successful business", positively correlating with the negotiation outcomes ($\tau b = -0.191$, $p < 0.05$), implying that business acumen reflecting non-gendered stereotypes was a positive mediator of negotiation outcomes. A point of convergence was drawn to determine whether the cumulative effect of predominant personality traits, negotiation skills, and gender-based stereotypes could effectively predict negotiation outcomes. The model effectively explained 42.6% of the variation in the negotiation outcomes by the independent variables ($\chi 2(8) = 44.79$, $p < 0.001$; goodness of fit $p = 0.05$), potentially endorsing the violation of gender-based stereotypes and predominant personality traits and negotiation skills as potent influencers of negotiation outcomes as per the EVT.

## 4. Discussion

Due to the COVID-19 pandemic, the Government of Mauritius imposed several movement restrictions in the country, the impact of which was observed in this study's response rate. Ideally, the response rate is critical to a study in relation to its internal validity and reliability, as observed by Mittelstaedt [56]. Despite the low response rate,

good internal validity, as supported by the high coefficients of Cronbach's Alpha, and good reliability, as per the pilot testing, were noted.

*4.1. The Relationship between Personality Traits and Negotiation Skills*

Corroborating the theory of situational psychology, most legal officers did not have a predominant personality trait; the dominance level of one trait is impacted by situational factors [57,58]. Nonetheless, the traits conscientiousness and neuroticism rated highest and lowest, respectively, among the participants. Interestingly, individuals "with high levels of conscientiousness are described as persistent, hardworking and self-disciplined" [59,60], while high neuroticism can be synonymous with low processing efficiency as well as poor performance in demanding situations [61,62]. Hence, this particular result is noteworthy as it aligns with the professional profile of the sample population.

Moreover, substantiating previous results, a significant negative association was observed between neuroticism and negotiation skills [63]. This inverse relationship between negotiation skills and high levels of neuroticism can be further explained by the theory of arousal whereby significant neurotic symptoms such as anxiety cause an increase in arousal level which hinders the concentration of the negotiator [64,65]. Barry and Fulmer [66] further argue that high levels of neuroticism are linked to a recollection of "more negative words and made more negative overall judgments" and negotiation requires quick strategic thinking focusing on the positive outcomes of the negotiation process. Hence, negotiators with the personality trait neuroticism can lead to poor negotiation outcomes.

*4.2. Gender-Based Stereotypes and Personality Traits among Legal Officers*

This study also demonstrates differences in conscientiousness, neuroticism, and agreeableness between males and females, a finding which can be corroborated by previous studies [67–69], whereby conscientiousness is believed to be characteristic of women, with cultural factors influencing the early development of conscientiousness in women [70]. As for the gender differences in neuroticism and agreeableness, this particular result can be aligned with the work of Costa and Terracciano [71] and Furnham and Cheng [72], respectively, with women showing higher levels of agreeableness as a result of socialization, while higher levels of neuroticism could be associated with their hormonal levels as well as their susceptibility to depressive disorders [73–75]. Of note, since both neuroticism and conscientiousness were more prominent in female legal officers, it can be assumed that conscientiousness countered the negative effect of neuroticism [76] given the good level of negotiation skills observed. In agreement with Edwards [77], the current findings denote that the majority of participants believed that gender should not influence the equality of opportunity to participate in management training programs nor discriminate between the work of males and females. Interestingly, supporting previous findings [78,79], most participants believed that women can handle their work pressure. Conversely, participants also believed that pregnancy can be a deterrent to further employment opportunities, substantiating the findings of Heiskanen and Rantalaiho [80] on how organizations unconsciously lay rules and practices that are more convenient to male employees who have fewer responsibilities compared to women and when the latter are unable to meet by deadlines, they are considered as less desirable for certain opportunities. Similarly, Choroszewicz [81] postulates that the equality of opportunity is a well-covered myth given that women themselves try to limit their opportunities to "provide them with greater flexibility to reconcile professional and family responsibilities".

Furthermore, a small percentage of female participants (≈39%) still felt that they were judged or shamed for not being feminine or masculine compared to approximately 15% of male participants. This particular finding further denotes a characteristic of symbolic violence, namely "a subtle and invisible form of dominance which is rarely identified as such by the individuals subject to it" [81], while Banchefsky and Westfall [82] identify this link among traditionally gender-specific professions, whereby certain professions such as educators and carers were suited to females with more feminine features while

those with less prominent femininity were identified as scientists. Hence, this particular finding can be attributed to the belief that the legal profession is profoundly believed to be male-dominated [83].

### 4.3. Link between Gender Stereotypes and Negotiation Skills

Additionally, as hypothesized, gender has an influence on the negotiation process, whereby almost 53% of participants, mostly females, were in agreement that their negotiation skills were influenced in a gender-dominated meeting. Interestingly, previous studies propose the theory of "stereotype threat" [35,84] to explain this link, whereby women develop expectations based on their opponents which then influence their negotiation skills while men only focus on negotiation outcomes.

Supporting the present finding that the majority of legal officers had good negotiation skills, Menkel-Meadow [85] argues that most researchers of psychology, economics, or even game theory use lawyers' knowledge of negotiation constructs to further their understanding. Hence, individuals with a legal background are expected to have a better understanding of negotiation and, thus, better negotiation skills. Moreover, corroborating previous studies, the present findings demonstrate that having good negotiation skills can impact the negotiation outcome [1]. Nonetheless, negotiation outcomes are also impacted by "the lawyer's authority, credibility, demeanor, and tactics" as observed by Wenke [86]. Accordingly, lawyers' creativity accounts for outcomes in the negotiation process [87], validating the findings that negotiation skills influence negotiation outcomes but only to a small extent.

### 4.4. Other Significant Variables Influencing Negotiation Skills

Further analyses were carried out to investigate whether profession, as well as years of experience in negotiation, influenced negotiation skills. A significant relationship was noted between negotiation skills and profession whereby counsels were seen to have better negotiation skills. Following the study of Deusen and Mark [88], this finding could be attributed to the fact that counsels tend to have a dual duty of both executive and attorney; thus, they are equipped with better negotiation skills. Interestingly, the number of years of experience in negotiation has no influence on the skills required to negotiate, which was further supported by the analysis of Williams [89] of the "lure of minimal competence" explaining how a negotiator's effectiveness is not influenced by years of experience [90] as individuals believed they had reached their optimal competence and stopped aiming higher over the years.

### 4.5. The Application of EVT and Negotiation Skills

The findings of the EVT were drawn upon in the present study through prescriptive gender stereotypes [91,92], whereby participants believed that there is a specific job for each gender. This study shows an expectancy violation with regard to male personality traits whereby most participants believed that men can be both caring and emotional. Given that this violation was significantly associated with excellent negotiation skills, it can be termed as a positive expectancy violation or even a pleasant surprise enhancing the negotiation, as per the EVT [2,40,41]. Kulik and Olekalns [2] further explain this finding in relation to personalized employment relationships in the workplace; women do not show many expectancy violations, fearing reprimands or negative evaluation [93], while men demonstrate positive expectancy violations. This study provides significant contributions to the EVT by extending its principles in the legal field through the novel link between personality traits, negotiating behavior, and positive or negative violations based on gender stereotypes. The results further provide an interesting outlook on the EVT, with the inclusion of how negotiating behaviors based on specific personality traits can act as positive and negative violations to further advance or limit negotiation outcomes. Hence, this study provides significant pointers for businesses on how to endorse more fruitful

negotiations with a greater focus on positive expectancy violations rather than negative ones and what personality traits could be further encouraged (Figure 1).

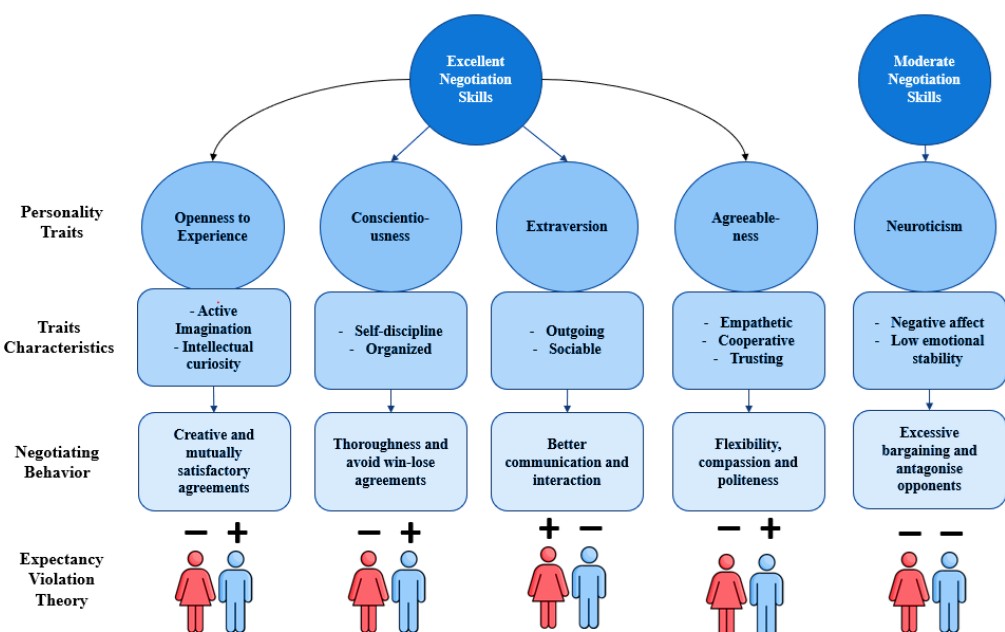

**Figure 1.** Conceptual model connecting negotiation skills, personality traits, and expectancy violation theory.

*4.6. Future Directions*

Further work regarding negotiation could take several directions given that negotiation is an intricate human interaction with multidimensional push and pull factors. Further research opportunities stemming from the present findings in the field of negotiation in Mauritius could prospect into dimensions such as culture, emotional intelligence, communication styles, and the lived experiences of the participants to further probe into the process of negotiation. Additionally, research on situational factors such as the negotiation topic, negotiating counterparts, and negotiating environment and their plausible influences exerted on negotiation skills and outcomes would provide fundamental information. Although gendered differences in terms of negotiation skills and outcomes were insignificant in the present study, the prevailing organizational factors such as a gendered predominant environment could be a latent variable to be considered in future studies.

**5. Conclusions**

The amalgamative effect of personality traits, more specifically, neuroticism, and gender on negotiation skills remain the focal finding of this study, corroborating previous studies. The findings further add significant support to existing studies in which negotiation outcomes were influenced by negotiation skills. Moreover, the use of the EVT further explains that gender stereotypes create expectations of feminine or masculine behaviors and elaborates on how violations of this expectation can be interpreted. Hence, given the prominence of good negotiation skills among most legal officers, it is important to accentuate the development of negotiation skills as part of legal studies. The development of negotiation skills is a critical part of most legal procedures if not all. For instance, as observed by Tyler and Cukier [94], "observation of experts, emotional intelligence and analogical reasoning" could be adopted while teaching negotiation in law schools. Additionally, to obtain seamless negotiation skills irrespective of gender, Recalde and Vesterlund [95] postulate that transparency is of utmost importance, as it shall help to equalize the process of negotiation for both males and females.

Furthermore, considering that neuroticism was seen as a deterrent to good negotiation skills and women were observed to score higher on the neuroticism level compared to

men, to ensure uniformity in negotiation skills among both genders, it is important to find solutions to manage the level of neuroticism among women. It is believed that neuroticism increases with age [96], while Lahey [97] argues that neuroticism is associated with poor mental health. Accordingly, to promote low levels of neuroticism for better negotiation skills, it is essential to sensitize legal officers on the importance of maintaining good mental health given the considerable pressure observed within the legal field. Thus, this study not only provides critical inputs on how to improve the negotiation processes for more constructive outcomes but also encourages readers and observers to think outside the box and question the underlying causes of high neuroticism as well as gender-based stereotypes.

**Supplementary Materials:** The following supporting information can be downloaded at https://www.mdpi.com/article/10.3390/businesses4010005/s1, Table S1: Demographic details of participants; Table S2: Comparative analysis between gender and personality traits; Table S3: Predominant personality traits depicted across Participants; Table S4: The perception of Women in equal or empowered roles in the workplace; Table S5: Stereotypical Gender Roles; Table S6: Mapping negotiation skills score against gender and profession; Table S7: Predominant personality traits across gender and negotiation skills; Table S8: Negotiation outcomes across predominant personality traits.

**Author Contributions:** Conceptualization, methodology, data curation, data analysis, and manuscript preparation, N.M. Supervision, K.A.-A. Manuscript preparation, editing, and review, M.P. and J.S. All authors have read and agreed to the published version of the manuscript.

**Funding:** This research received no external funding.

**Institutional Review Board Statement:** The ethical considerations of this study were reviewed by the Faculty of Law Dissertation Committee, University of Mauritius.

**Informed Consent Statement:** Informed consent was obtained from all subjects involved in this study.

**Data Availability Statement:** The authors confirm that the data supporting the findings of this study are available within the article and its supplementary materials.

**Acknowledgments:** The authors wish to thank colleagues from the Faculty of Law and Management, University of Mauritius, who provided insight and expertise that greatly assisted this research.

**Conflicts of Interest:** The authors declare no conflict of interest.

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
