# Peer review of "Mapping Personality Traits and Gender-Based Stereotypes on Perceived Negotiation Skills"

_2673-7116, doi:10.3390/businesses4010005_

Round 1

Reviewer 1 Report

Comments and Suggestions for Authors

This is an important piece of work being reported through this paper. At a general level, the paper has clear objectives, with needed alignment to the theory, methods, and analysis of the work. The use of expectancy theory through gendered lenses is certainly commendable as it provides a unique set of lenses through which one can frame the understanding of negotiation skills. The inclusion of gender-sensitive question as a sub-measure within the instrument is also commendable as it goes beyond the broader tendency for the single inclusion of a question on sex, as a proxy for gender. The attempt to test for quantitative measures of this gendered sensitivities/disposition also adds to the relevance and contribution of this paper. While this is deduced, this is not clearly articulated by the author(s).

However, to improve the quality of the paper however there are a few recommendations:

1. Extend the discussion to speak to the specific contributions that are being made through the paper. The discussion can also expand the section on the implications for theory and practice to show how this gendered framing of expectancy theory can be extended through theoretical revisions and methodological explorations.

2. Include a visual that demonstrates how the theory aligns with the measures being tested through this study.  This visualization will help the reader to see the connections that are being made through this work.

3. Strengthen the introduction of the paper to signal the use of gender and testing of its relevance to negotiation skills and to locate this within the gaps for the field.  Author(s) can also consider the inclusion of more expansive research/literature that speaks to the relevance of gender in relation to negotiation skills.

4. Provide greater details on the methodology for this work including the research design (rationale and type), the process of instrument construction and verification, the process for data collection, sampling, and establishing reliability. This is necessary to establish the rigor of the process and the overall merit of the paper.

Comments on the Quality of English Language

None

Reviewer 2 Report

Comments and Suggestions for Authors

1. Abstract

- Population and sample determination need to be added

- Add the managerial and theory contribution

2. Keywords : write alphabetically

3. Introduction

- Phenomenon theoretically taking into account differences of opinion between researchers, and the position of this research is not clear.

-. The practical phenomenon of why research results are important has not been explained clearly

-. Need to add structure to article writing

4. Add literature review

-. It is necessary to provide additional relationships between concepts to fit the sub section 3.4. and 3.5.

-. It is necessary to add a research model in the form of figure to make it easier for readers to understand.

-. hypotheses need to be established for this research to fit the discussions

5. The table 1, title is inappropriate because there is an alpha value for all variable

6.  point 3.4 Data needs to be made in table form, to make it easier to read the contents of the journal

7. point 3.5. Data needs to be made in table form, to make it easier to read the contents of the journal

8. End of discussions : need to add managerial implication and theory contribution

Comments on the Quality of English Language

Need to improve 

Reviewer 3 Report

Comments and Suggestions for Authors

Thanks for letting me review a paper entitled “Mapping Personality Traits and Gender-Based Stereotypes On Perceived Negotiation Skills”

The paper may be published after considering the below major issues:

1-      The topic is good, however the research problem and motivation are not clearly identified and justified.

2-      sample size of 270 need justification for its adequacy.

3-      Data were analyzed on SPSS version 21.0., outdated data analysis techniques which failed to infer causality or test a complexed mode with multiple dependent and independent laten variables (try conducting PLS-SEM)

4-      I need to see the tables of analysis in the appendix to evaluate the results.

5-      Theoretical and practical implications need more effort and justifications.

6-      No limitation or further study opportunities  

Comments on the Quality of English Language

Minor 

Round 2

Reviewer 1 Report

Comments and Suggestions for Authors

This is a significantly improved paper with clear evidence that the authors have taken the time to address the suggestions of the reviewers. The paper now reads well with a minor request to add information on the process for stratifying the sample. This would give the reader a sense of the process and a better understanding of how the researchers/authors executed this type of work. In the discussion, authors can also speak to how the theory or perspective can be further strengthened to address some of the findings of the paper.  

Author Response

  1. The methodology part was further edited to explain the nature of the stratification applied in the context of the research, the latter which can be read as ‘The inclusion criterion of the study population was being a legal officer having at least six months of experience in the negotiation field. Furthermore, given the ratio of barristers to attorneys (5:1) within the Mauritian legal system. The sample size, 270, was calculated using Slovin’s formula accounting for a 95% confidence interval and a 5% margin of error, based on a population size of 899 as per records of the Mauritius Bar Council. It was estimated that at least 108 barristers would be required, accounting for a response distribution of 50%. Given the plurality of the roles undertaken by legal officers, segregation according to distinct roles was minimized while the focus was mainly on precursor negotiation experience’ at Pg 3-4, L137-145 in the revised manuscript
  2. Amendments with regard to the theory and perspective have been made which can be read as, “The results further provide an interesting outlook of the EVT, with the inclusion of how negotiating behaviors based on specific personality traits can act as positive and negative violations to further advance or limit negotiation outcomes. Hence, the study provides significant pointers for businesses on how to endorse more fruitful negotiations with a greater focus on positive expectancy violations rather than negative ones and what personality traits could be further encouraged (Figure 1)” at Pg 10, L 426-432 in the revised manuscript. 

Reviewer 3 Report

Comments and Suggestions for Authors

can accept the paper in its current form

Best wishes 

Comments on the Quality of English Language

minor 

Author Response

We thank the reviewer for this recurrent comment. The manuscript has been proofread and reworked to improve the quality of the English language.